# The Genetics of Intellectual Disability

**DOI:** 10.3390/brainsci13020231

**Published:** 2023-01-30

**Authors:** Sandra Jansen, Lisenka E. L. M. Vissers, Bert B. A. de Vries

**Affiliations:** Department of Human Genetics, Donders Institute for Brain, Cognition and Behaviour, Radboud University Medical Center, P.O. Box 9101, 6500 HB Nijmegen, The Netherlands

**Keywords:** intellectual disability, genetics, phenotype, genotype, next-generation sequencing

## Abstract

Intellectual disability (ID) has a prevalence of ~2–3% in the general population, having a large societal impact. The underlying cause of ID is largely of genetic origin; however, identifying this genetic cause has in the past often led to long diagnostic Odysseys. Over the past decades, improvements in genetic diagnostic technologies and strategies have led to these causes being more and more detectable: from cytogenetic analysis in 1959, we moved in the first decade of the 21st century from genomic microarrays with a diagnostic yield of ~20% to next-generation sequencing platforms with a yield of up to 60%. In this review, we discuss these various developments, as well as their associated challenges and implications for the field of ID, which highlight the revolutionizing shift in clinical practice from a phenotype-first into genotype-first approach.

## 1. Introduction

The DSM−5, developed by the American Psychiatric Association, describes intellectual disability (ID) as a defect in intellectual functioning and adaptive behavior that starts in the developmental period and influences three domains in daily life: firstly, the conceptual domain, which includes knowledge, reasoning, memory and the ability to write, read and do math; secondly, the social domain, which describes functioning in social interactions such as maintaining friendships, communication skills and empathy; and thirdly, the practical domain, which includes personal care, organizing daily life, the ability to attend school, have a job and manage finances [1]. Severity has been categorized into mild, moderate, severe and profound and is based on adaptive functioning in the DSM−5 [1]. Although IQ scores were excluded from the DSM−5, the corresponding IQ scores from the DSM−4 (55–70 (mild), 40–55 (moderate), 25–40 (severe) and <25 (profound)) are still widely used. Global developmental delay (DD) in children under the age of five years can be a predictor for ID, but some of these children attend regular education later in life [2,3]. Individuals with an IQ of 70–85 have borderline normal intelligence, which is not considered intellectual disability [3].

The prevalence of ID (e.g., IQ < 70) is ~2–3% of the global population (Figure 1). In low-income countries, this is higher due to less access to healthcare facilities and prevention programs, which increases the risk of ID through environmental factors [4]. Having a child with a neurodevelopmental disorder (NDD), of which ID is a subgroup, has a great impact on parents regarding many aspects of daily life [5]. In addition, the financial burden on society is huge, and the healthcare needed by individuals with ID accounts for ~9% of the total health care costs in The Netherlands [6].

ID is both clinically and genetically heterogeneous (Figure 2). Clinical heterogeneity is reflected in the diversity of different ID disorders but also by the clinical variability in a specific disorder caused by mutations in the same gene [8,9]. Non-syndromic ID, or “pure ID”, is usually referred to if ID is the only feature observed in the patient, whereas syndromic ID is defined as having ID with one or more co-morbidities or facial dysmorphisms [10]. However, in clinical practice, it is difficult to rule out mild dysmorphic findings or subtle neuropsychiatric abnormalities that are often seen in individuals with ID. In addition, the distinction between syndromic and non-syndromic ID is not black and white, and there is debate when non-syndromic ID turns into syndromic ID, which makes the distinction rather subjective [10,11,12]. The co-morbidities that are frequently seen in individuals with ID comprise congenital abnormalities and other neurodevelopmental disorders (NDDs), such as epilepsy, autism spectrum disorders (ASD) and attention deficit hyperactivity disorder (ADHD) [13]. These NDDs are individually also rather common with the prevalence of ASD being ~1.7% of school-aged children in the US, a life-time prevalence of epilepsy of ~0.8% and a prevalence of ADHD of 5.3–7.2% [14,15,16,17]. The genetic heterogeneity is reflected by the fact that already ~800 genes were reported to be involved in ID in 2016, but the latest ID gene panels used in diagnostic laboratories contain >1500 genes. Moreover, estimations predict that ~10% of all 20,000 human genes might be involved [18,19]. Additionally, there is a large genetic overlap between ID, ASD and epilepsy [20,21,22].

The clinical and genetic heterogeneity complicates the process of obtaining a precise clinical and genetic diagnosis which is important for the affected individual, parents, relatives and other caregivers. Having a diagnosis means it is possible to make a management plan based on the available knowledge on the specific disorder. Management currently focuses on the surveillance and treatment of symptoms, such as physiotherapy, speech therapy or anti-epileptic drugs in the case of epilepsy. Moreover, a correct diagnosis allows patients and families to connect with other families, self-help groups or advocacy groups. In addition, families might gain access to special, condition-specific support programs. Treatment of cognitive impairment as such has been the focus of research projects [23] but is not available, as of yet, except for several metabolic disorders [24]. Before pharmacological treatment can be developed, the pathophysiological mechanism of the specific ID disorder, hence the genomic defect or compromised pathway, needs to be defined. Additionally, having a diagnosis can help the parents and other caregivers to better understand why the disorder occurred, assist in accepting that this happened and provide crucial information on the prognosis of the disorder and the recurrence risk [25,26,27].

To get to a diagnosis, two main approaches are available: “phenotype-first” and “genotype-first” (Figure 2). Due to novel diagnostic advancements in recent years, the “genotype-first approach” has become more and more the first step within the diagnostic process in ID.

## 2. Phenotype-First Approach

Phenotyping in clinical genetic practice means obtaining a detailed medical history and performing a thorough physical examination, including a detailed assessment of the dysmorphology. Although strictly speaking not part of the patient’s phenotype, a detailed family history is also obtained. In individuals with intellectual disability, additional investigations are usually also performed, such as electroencephalography (EEG), magnetic resonance imaging (MRI) of the brain and metabolic screening. The clinical work-up is followed by making a differential diagnosis: when patients fulfill certain criteria, a clinical diagnosis is accomplished. Specific genetic testing is requested if the gene is known. This phenotype-first approach has been relatively successful in known, clinically recognizable, single-gene disorders and some microdeletion/duplication syndromes. For a long time, a clinical diagnosis was the only option for several syndromes as a molecular confirmation was simply not feasible. Nice examples are Williams syndrome, clinically described in 1961 by Williams et al. [28], for which the underlying the molecular defect, a microdeletion on chromosome 7q11.23, was identified 32 years later [29,30] and Noonan syndrome, first reported in 1968 [31], followed by the identification of the causative *PTPN11* gene in 2001 [32].

However, even after the identification of the molecular defect, obtaining a diagnosis still depended, to a large extent, on the clinician’s expertise and ability to recognize the many hundreds of clinically recognizable disorders. Moreover, the approach fails when the patient’s characteristics do not fit a known disorder, the syndrome is genetically heterogeneous or the genetic defect is unknown. In these latter scenarios, the clinician simply does not know which gene to test, and a clinical diagnosis cannot be confirmed by any (molecular) testing.

Importantly, in the second half of the past century, the only way to identify a causative gene mutation was using Sanger sequencing (Figure 3), a technique developed by Fred Sanger in 1977. This first-generation sequencing required the DNA to be amplified using PCR, before sequencing by gel electrophoresis was performed. As Sanger sequencing only allowed for the testing of one gene at a time, sequential single-by-gene testing was often performed until a causative mutation was found based on clinical suspicion. This diagnostic process often took years, was a burden to the patients and families involved and is often referred to as the “diagnostic odyssey” [33]. Moreover, for diagnosing patients with more than one Mendelian disorder, the phenotype-first approach would not work. In addition, this type of genetic testing was relatively expensive, as it required many manual laboratory and interpretation steps [33,34,35,36]. Moreover, in only 7% of patients could the gene mutation causing the phenotype be identified. This low diagnostic yield led to a search for a genome-wide screening test without the a priori knowledge of the genomic locus to be involved, yet more successful and faster than Sanger sequencing.

## 3. Genotype-First Approach

Although early chromosomal analysis using karyotyping was performed in individuals suspected of a specific disorder, such as trisomy 21, and, thus, was a phenotype-first approach, later, karyotyping was used more broadly in individuals with ID and congenital abnormalities (Figure 2) [37,38]. It can, therefore, be regarded as the earliest genotype-first method and was successful in detecting large chromosomal rearrangements (>5–10 Mb) which explain 10–15% of ID cases. In addition, the discovery of the fragile X chromosome in the early 1970s gave an extra diagnostic yield, with ~1% of the males with intellectual disability being affected [39]. It also opened the way to linkage studies in large X-linked families with ID, and subsequently, the identification of over 140 genes on the X-chromosome involved with ID [40,41]. In the 1980s, fluorescence in situ hybridization (FISH) was developed, which could detect smaller copy number variations (CNVs) than karyotyping and was used to diagnose some of the “old” classical syndromes such as Williams syndrome and Rubinstein–Taybi syndrome in the 1990s [30,42]. After the publication of Flint et al. in 1995 on subtelomeric abnormalities in relation to ID [43], the multiprobe FISH was shown to be highly suitable for detecting those subtelomeric deletions and duplications which accounted for 2.5–5% of the diagnostic yield in individuals with ID [44,45,46].

Subsequently, in the first years of the 21st century, multiplex ligation-dependent probe amplification (MLPA) and genomic microarrays, including both microarray-based comparative genomic hybridization (CGH array) and single nucleotide polymorphism genotyping array (SNP array), were developed (Figure 3). These techniques were used to identify CNVs throughout the genome, increasing the diagnostic yield by another ~15% [18,47] and are still widely used to diagnose ID. The introduction also led to the identification of novel ID syndromes such as 1q21.1 microdeletion syndrome, 15q24 microdeletion syndrome and the 17q21.3 microdeletion syndrome, also known as Koolen–DeVries syndrome [48,49,50]. All are nice examples of the genotype-first approach after a general molecular screen leading to a clinical characterization of a novel syndrome and sometimes even to a causative single gene.

## 4. The Era of Next-Generation Sequencing

While chromosome microarrays enabled the genome-wide unbiased detection of structural rearrangements, similar methods for sequence variants at base-pair resolution were still lacking. Next-generation sequencing, referring to the massive parallel sequencing methods which were established at the beginning of the 21st century, promised to overcome the technical challenge of sequencing all genes, or even the whole genome, at a base-pair level after a period in which Sanger sequencing dominated [51]. The term NGS includes all technologies that use massive parallel sequencing, but it comprises multiple different methods and platforms (Figure 4). Importantly, they all overcome the challenges associated with clinical and genetic heterogenetic disorders, facilitating the ultimate genotype-first conditions. NGS can make use of short and long reads. Short-read platforms are very popular because of the high number of reads produced in a run, but long-read platforms are better at detecting structural variation and overcoming problems due to less characterized regions in the human genome [52,53].

Within a diagnostic setting, whole exome sequencing (WES) has been introduced over recent years for several disorders, including ID-to-screen for coding variants of known genes [54,55]. An important challenge is the interpretation of the ~20,000 protein-coding variants that are found in, for instance, the WES data of a child with ID, where any one of those may cause a disorder [56]. Bioinformatic pipelines are crucial in the process to prioritize these to find the clinically relevant ones. In this process, first, variants that do not fulfill quality standards are filtered out [57]. Secondly, variants that are frequently observed in the general (healthy) population and are suspected to be benign are also filtered out. For this step, large datasets of control populations are used, such as dbSNP [58], the 1000 Genomes Project [59], GoNL [60] and gnomAD [61] (successor of ExAC [62]). Next, variants are prioritized based on the gene of interest, type of mutation, its impact on protein function and allelic composition to fit the expected inheritance pattern. For the interpretation of the functional impact of the variant, many in silico prediction programs and scores have been developed, such as CADD, SIFT, PolyPhen, PhyloP and MutationTaster [63]. These tools make use of the conservation of nucleotides and amino acids, biochemical characteristics of amino acids, effect on splicing and more, to predict pathogenicity. This strategy, on average, leads to 5–10 variants per individual that require an extensive genotype–phenotype assessment [56]. Of note, as severe ID is associated with reproductive lethality, de novo mutations, identifiable from a trio-based sequencing approach [18,56,64,65,66,67,68,69], can be used as additional prioritization step as these have been shown to be an important cause. Detailed genotype–phenotype studies, in turn, include searching the literature and disorder databases to find out whether or not the variant or gene is already known to be related to ID or other NDDs. If a causal relation has not been proven yet, the expected effect on the phenotype can be corroborated using combined data on the protein function as described in the literature or by performing functional analyses (Figure 5).

WES has been shown to be successful in individuals with ID, ASD and epilepsy [70,71,72,73] in both research and diagnostic settings. Overall, the use of WES in many cohorts of individuals with a variety of (neurodevelopmental) disorders, including ID, has been shown to provide an overall diagnostic yield of WES in individuals with ID of ~30% (Figure 6) [18,56,64,65,66,67,68,69]. Due to the improvement of diagnostic yield over the pre-NGS era used technologies, as well as the turnaround time and reduction in NGS costs, WES has become a first-tier test in most developed countries [27,34].

## 5. Gene Discovery in ID

Gene discovery in ID has been following the development of new techniques in genetics (Figure 6). It started with the identification of chromosomal anomalies such as trisomy 21 and a “fragile site” at the X chromosome in an X-linked pedigree which was later known as Fragile X syndrome and turned out to be caused by an expansion of trinucleotide repeats [74,75]. As ID was more prevalent in males than in females and the search for causative genes in large pedigrees facilitated gene discovery, researchers in the 1990s focused on X-linked ID. This led to the identification of more than 100 X-linked ID genes, which account for 5–10% of ID in males [76]. In families with more than one affected child and in populations where consanguine relations are common, autosomal recessive mutations were suspected and identified [77,78]. These account for 37 to 90% of ID in consanguineous families [78], but these families only account for a small percentage of the total patient group.

Family history has not only been helpful in the gene discovery of X-linked and autosomal recessive disorders. The absence of a positive family history is indicative as well as it has long been recognized that there is a high heritability in common mental illnesses, including ID. This was based on the observation that mental illnesses occurred more often in family members of an affected person and on twin studies [79]. On the other hand, mental illnesses often develop in childhood, and individuals with a mental illness, especially ID, often do not reproduce in line with negative selection pressure [79]. It was thought that either a polygenic model or recently originated rare mutations in functional relevant genes would underlie this paradox. The detection of CNVs in individuals with ID and ASD that were not found in their parents and, thus, had arisen “*de novo*” was the first confirmation of the latter hypothesis [22,47,80,81,82]. This was further confirmed when de novo single nucleotide variants (SNVs) were also identified in individuals with ID and other NDDs since 2010 [56,64,65,67,68,69,70,83,84,85,86]. This has led to the identification of numerous novel genes involved in ID (Figure 6) and subsequently, several novel ID syndromes such as the Jansen-DeVries syndrome (OMIM#617450) and the Zhu–Tokita–Takenouchi–Kim syndrome (OMIM#617140) (Figure 5).

For most of these novel syndromes, the process to their first description is identical (Figure 5): after the identification of the first patient with NDD, a search for a larger series starts. This can be either through (inter)national collaborations and matchmaker exchange programs or, alternatively, by the active (re)sequencing of patients with a similar NDD phenotype using targeted enrichment. For the latter, a popular approach has been the use of molecular inversion probes (MIPs) [22,87,88,89,90], which is a high throughput and affordable technology to effectively sequence a large cohort of individuals on pathogenic variants in 5–50 genes simultaneously within a single experiment. Once multiple patients have been identified, more detailed further clinical characterization is performed using deep-phenotyping to established the phenotypic spectrum. Subsequently, depending on the function of the protein, functional testing can be developed to prove a deleterious effect of the various gene variants. In addition, various disease models, either animal or cellular models, can contribute to the further molecular characterization of the gene/protein and its pathophysiological mechanism.

## 6. From Exomes to Genomes—From Coding to Non-Coding Variants

Without a doubt, the use of WES has changed the diagnostic and research potential for identifying the molecular causes of disease. However, given that the majority of patients still remain undiagnosed after WES, whereas a genetic cause was expected, it has already been long hypothesized that either the tests failed to identify the pathogenic variant or our ability to recognize the pathogenicity of the variant was insufficient. From a technological point of view, progress was made with the introduction of (short-read) whole genome sequencing, which eliminated the enrichment step for the protein-coding sequence (Figure 7) [70]. This not only allowed for the identification of non-coding variants but also provided more uniform coverage throughout the genome which is useful for the identification of structural variation at base-pair resolution. Comparative studies assessing the increase in the diagnostic yield for NDD in (short-read) WGS over WES have shown that—although a few additional diagnoses are established—there is not a significant increase in diagnostic yield when only replacing WES by WGS to interpret the variants affecting the protein-coding sequence [57]. Similarly, the number of additional diagnoses by “simply” interpreting non-coding variants has not been highly successful as interpretation rules for protein coding variants are not easily transferred to non-coding variants [58,70,72]. Despite these challenges, some studies on non-coding variation in ASD showed enrichment for certain de novo mutations or multiple de novo mutations in individuals with ASD [91,92,93], while others could not [94]. A recent study in individuals with NDDs, including ID, focused on three classes of regulatory elements and showed that de novo mutations in the conserved non-coding elements (CNEs) involved in the developing fetal brain were enriched in exome-negative patients. However, no single element reached statistical significance and could be regarded as causative for NDD [95]. The authors calculated that pathogenic de novo non-coding mutations probably account for less than 5% of exome-negative NDD patients, and it will take WGS in tens to hundreds or thousands of patient–parent trios to accomplish this.

The most promising interpretation strategies have so far been hypothesis-driven approaches rather than searching for statistical enrichment in non-coding sequence. For SNVs, these included the identification of non-coding variants affecting the so-called gene body, where rare (de novo) variants affected, for instance, sequences of the UTRs disrupting the wild-type translation start site [96,97] or (de novo) variants in introns, where they introduced cryptic splice sites leading to abnormal splicing and the inclusion of poison exons [98,99,100]. For non-coding structural variants, the molecular mechanism leading to disease has mainly been the dysregulation of spatial 3D-organization [101] and gene regulation, for instance, through the disruption of topological-associated domain structures [102,103,104].

## 7. Genomes: From Short Reads to Long Reads and The Incorporation of Other -Omics

Although only a limited number of non-coding variants from short-read WGS are currently being systematically interpreted due to a knowledge gap on recognizing pathogenicity, technology keeps on making progress from which the genetics landscape of NDD might be directly benefitting from. These opportunities include, amongst others, long-read whole genome sequencing and optical genome mapping for better variant detection (Figure 7) but also the use of other -omics technologies, such as transcriptomics and/or metabolomics, to complement variant interpretation.

Long-read WGS (LR-WGS) is a sequencing-based approach, considered to be part of the third-generation sequencers. The main advantage of LR-WGS is the length of reads, extending up to 10–50 kb depending on the technology, allowing the spanning of more complex regions of the genome during read mapping, and, thus, variant calling in regions previously inaccessible by short-read technology [105]. Ample examples have been published showing how LR-WGS applications were able to detect a disease-causing variant in patients with Mendelian disorders who had remained undiagnosed by standard short-read technologies [106,107]. Yet, more systematic studies revealed that LR-WGS was able to capture variants in ~35 Mb of the sequence that was inaccessible before, and that, more importantly, these regions contain >60 genes known to cause Mendelian disorders, including NDD [108]. In addition, others showed that the use of genomes from long-read sequencing inherently improve de novo mutation detection [109], and this increases the identification of such causal variants in NDD.

In contrast to LR-WGS, optical genome mapping (OGM) is a non-sequencing based technology, which is based on ultra-high molecular weight DNA molecules that are labeled at a 6-mer motif (CTTAG), allowing for high-resolution genome-wide structural variant detection [110]. The molecules are 150 kb to Mbs in length, providing an even better resolution for structural variant detection. Its use is mostly seen as a molecular replacement for routine cytogenetic assays and as such, fills a caveat that remained present after the introduction of short- and long-read sequencing [111]. Indeed, the technology has already demonstrated its use through the identification of disease-causing variants that had remained unidentified until the use of OGM [112,113].

In addition to reading the human DNA sequence, other -omics technologies are also finding their way into the clinical arena, of which transcriptomics and epigenomics have made the most headway. Initially, transcriptomics were applied to patients with rare (neuro)muscular and mitochondrial disorders as direct access to the affected tissue could be obtained for studies [114,115]. The use of transcriptomics in patients in whom exome interpretation alone did not identify the genetic cause yielded an additional 10–35% diagnoses. The first systematic transcriptomes for patients with neurodevelopmental disorders have also booked their first successes as was recently shown by analyzing 30 patients and obtaining a conclusive diagnosis in 27% [116]. The next challenge in the use of transcriptomics will be the upgrade from short-read sequencing technologies to long-read technologies (IsoSeq), allowing the interrogation of full native transcripts [117] as well as single cell transcriptomics [118].

For neurodevelopmental disorders, it has been well-established that the genes involved in the epigenetic machinery play an important role [119,120,121]. Notably, this includes imprinting disorders associated with ID, for instance, Prader–Willi (OMIM #176270) and Angelman syndrome (OMIM #105830). In recent years, it has also been increasingly observed that the disruption of genes involved epigenetic processes leave distinct, recognizable signatures, also referred to as epi-signatures [122,123,124]. These signatures are now not only steadily used as supplements to the interpretation of exome and genome sequencing data to help better understand the clinical relevance of the variants of unknown significance (VUS) in well-known ID genes [125] but also to enhance genotype–phenotype correlations [126,127].

## 8. Quantitative Phenotyping

The implementation of high-throughput sequencing technologies, such as exome sequencing and genome sequencing, have facilitated the ability to detect individuals with increasingly rare novel disorders; this also calls for novel methods for collecting phenotypic information. One of these novel tools is “quantitative facial phenotyping” or “facial recognition” (Figure 8). Facial recognition and matching algorithms have matured in recent years and can be exploited in a clinical setting to automate and objectively measure a person’s dysmorphology with minimal impact on the person [128,129,130,131]. The advantages of these methods are that they also include subtle features that may be difficult for clinicians to identify or recognize features that may not necessarily be considered dysmorphic. Moreover, they can be used to identify additional individuals with a known syndrome or aid the characterization of novel ID syndromes. Alternatively, they can support the interpretation of variants for which the functional impact may be uncertain, such as missense variants in a gene known to cause a disorder when mutated but for which clinical presentation is variable and/or includes a broad clinical spectrum.

The current state-of-the-art model is DeepGestalt [131,132], demonstrating 60% accuracy in 216 different genetic syndromes. DeepGestalt, however, similar to all other published studies since Ferry et al. in 2014, depends on proprietary, closed-source algorithms. With open science gaining traction in academia, the sharing of code, models and data (when possible) is desirable to ensure academic progress is not hindered and ensure anyone can use these models, verify them and further build on and improve them. A novel AI-based phenomics approach has recently been presented by Dingemans et al. who developed a PhenoScore based on the combination of facial recognition technology with human phenotype ontology (HPO) data analysis [133]. This novel open-science approach, leading to this PhenoScore, allows not only to quantify rare disorders but genetic variation as well [133].

## 9. Concluding Remarks

Since the identification of the double helix in 1953 and the exact number of the chromosomes in 1955, the diagnostic possibilities in ID have increased enormously. The ability to find the underlying molecular defect has increased in the NGS era and depending on the clinical preselection of patients, the diagnostic yield may be as high as 60%. These novel technologies in genomic analyses initiated the genotype-first approach and the finding of the de novo occurrence of genetic variants as an important cause of ID. With WES and WGS analyses having become routine and relatively easy procedures, the assessment of the phenotype in establishing the significance of the genomic variants has become the next essential challenge. To achieve this, novel tools will need to be developed varying from gene-specific functional tests and animal models to quantitative (facial) phenotyping. Ultimately, this will need to be followed by the next step: improving the treatment of patients with ID and its co-morbidities.

## Figures and Tables

**Figure 1 brainsci-13-00231-f001:**
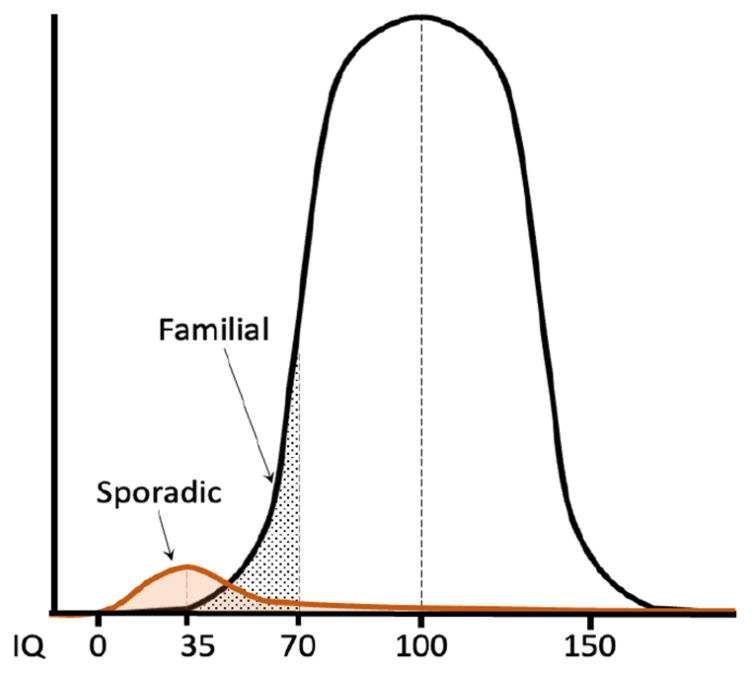
The IQ distribution of the “normal” population has the shape of a Gaussian curve, with the mean IQ set at 100 IQ points. IQ scores below 70 are observed in ~2.5% of the population. The second curve with the mean around 35 IQ points represents individuals with a pathophysiological cause underlying low IQ measurements (adapted from Zigler et al., 1967 [7]).

**Figure 2 brainsci-13-00231-f002:**
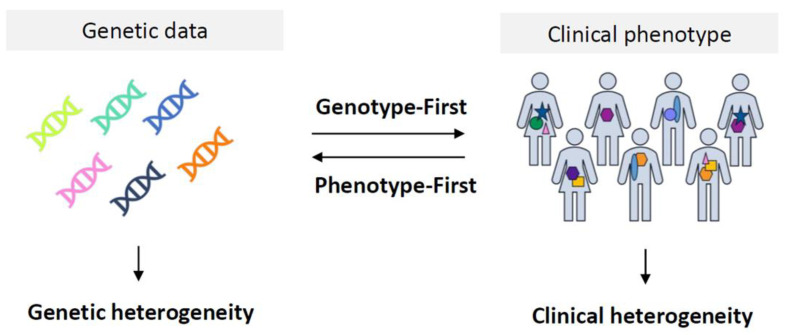
Graphical representation of the various concepts in the field of medical genetics. Genetic heterogeneity refers to the concept that pathogenic variants in multiple genes may lead to the same phenotypic presentation, whereas clinical heterogeneity refers to the fact that pathogenic variants in the same gene may lead to different clinical representations. A genotype-first approach starts from the genetic data point of view: after the identification of a pathogenic variant, the phenotype is thoroughly examined. In a phenotype-first approach, the clinical presentation of the patient with ID is obtained first, which is subsequently used as a guide for (targeted) genetic testing.

**Figure 3 brainsci-13-00231-f003:**
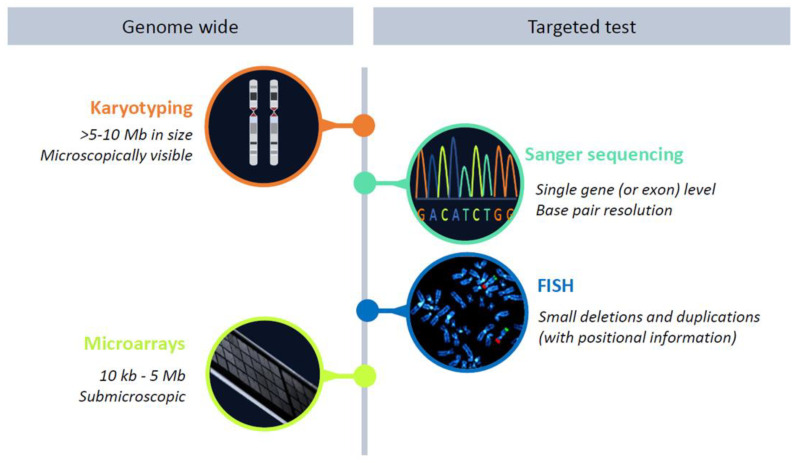
Schematic overview of the most commonly used cytogenetic and molecular genetic assays employed to elucidate the underlying genetic cause of ID in the pre-NGS era. Of note, genome-wide genetic assays can be used in both a genotype- and phenotype-first approach, whereas the targeted assays are mostly only used in a phenotype-first approach.

**Figure 4 brainsci-13-00231-f004:**
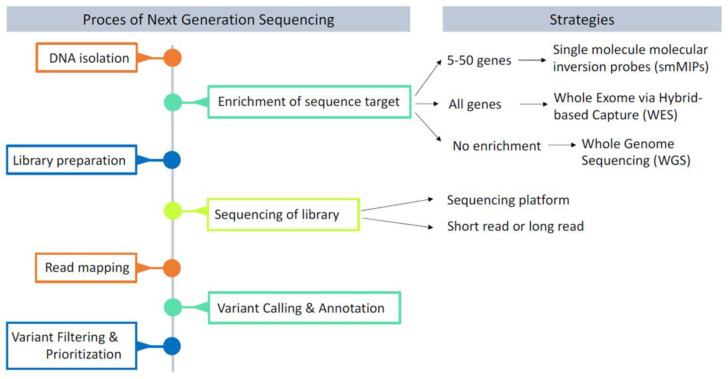
Stepwise delineation of the next-generation sequencing approach.

**Figure 5 brainsci-13-00231-f005:**
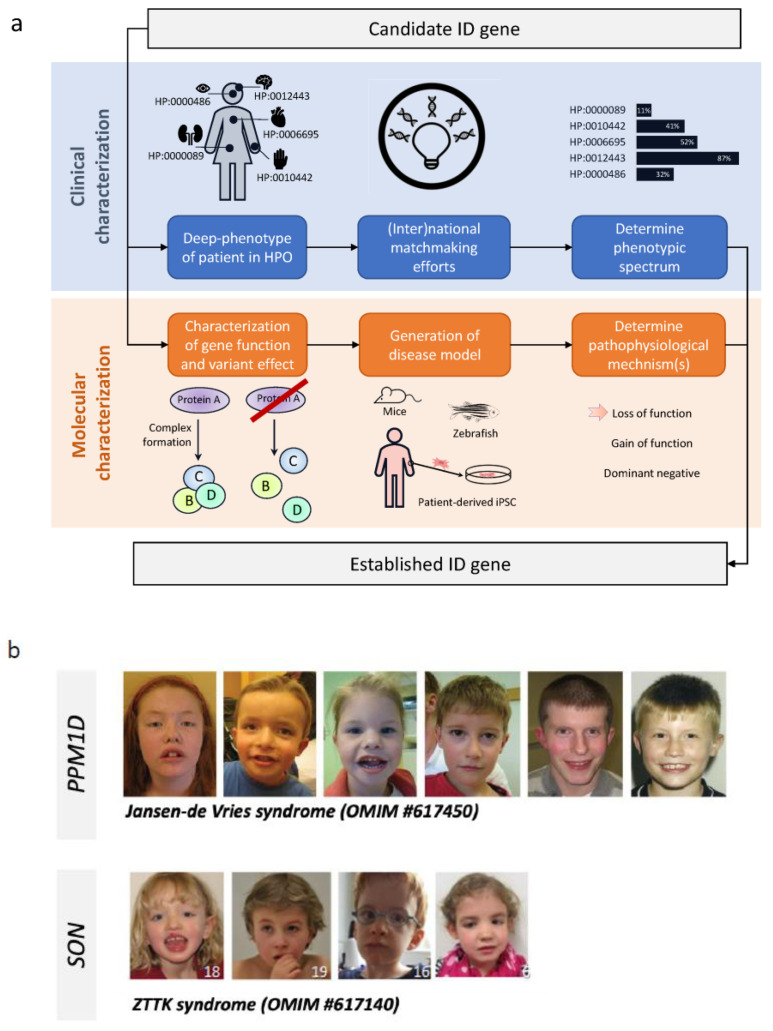
Schematic overview to reclassify a candidate ID gene to an established ID gene, with two successful examples. (**a**) The use of exome sequencing facilitates the identification of novel disease–gene associations. At first, a single patient is identified to have a (likely) pathogenic variant in a gene not previously associated with the disorder. Two parallel tracks are then started: one focusing on delineating the phenotypic spectrum of patients with (likely) pathogenic variants in the candidate ID gene and one focusing on the delineation of the molecular aspects of gene (dys)function. (**b**) Facial photos of two NDD syndromes identified through the use of exome sequencing: *PPM1D* involved in Jansen-de Vries syndrome, and *SON* in ZTTK syndrome.

**Figure 6 brainsci-13-00231-f006:**
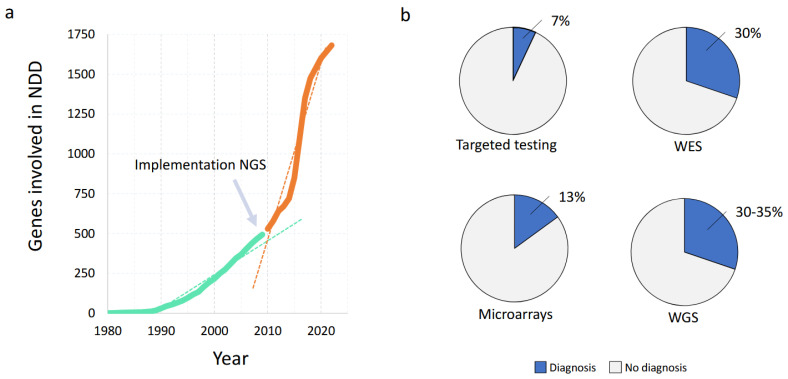
Schematic representation of the increase in genes associated with neurodevelopmental disorders and the respective diagnostic yields from assay used to find the underlying genetic cause. (**a**) The number of genes has significantly increased since the use of NGS-based assays (green line: pre-NGS; orange line: post-NGS, with dashed lines showing the trendline of the increase). (**b**) Diagnostic yields of the individual techniques when used as singular approach. A targeted strategy provides only a diagnosis in 7% of patients (phenotype-first), whereas the broader genotype-first approaches provide a diagnostic yield of up to ±35%.

**Figure 7 brainsci-13-00231-f007:**
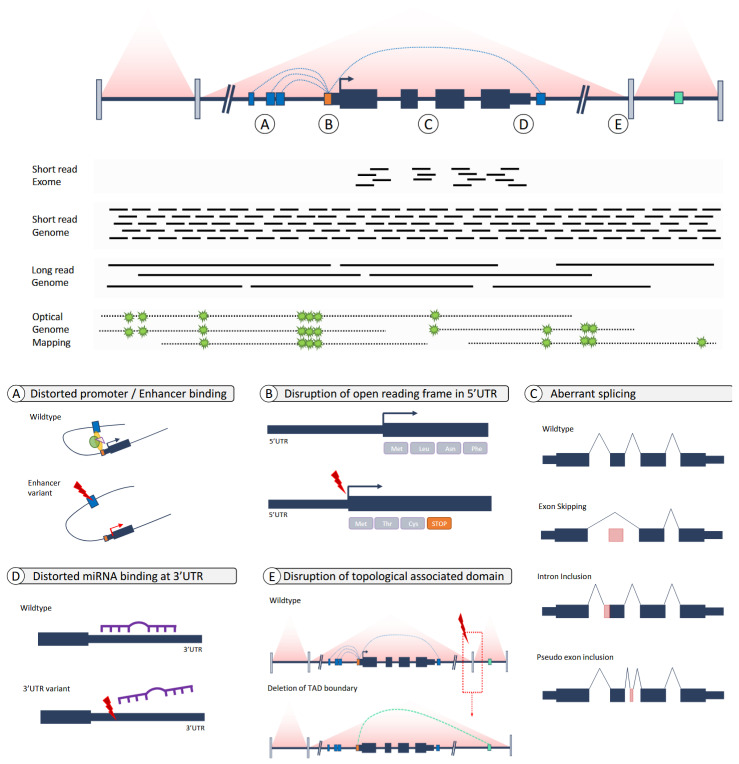
Overview of technological developments and the role of non-coding DNA variants in ID. Top panel shows schematic genomic structure with a gene in a topological-associated domain. Below this structure, different technologies are depicted which are commonly used to identify the genetic cause of NDD, including exome sequencing, short- and long-read genome sequencing and optical genome mapping (in green: labels detected in optical genome mapping technology). In the structure, the letters represent mechanisms, detailed out in the bottom panel, on how variants in non-coding DNA sequence can lead to disease. (**A**) Rare variants in an enhancer may lead to the distortion of binding of the enhancer to the promoter, effectively resulting in failure of transcription. (**B**) Rare variants in the 5′UTR may lead to the introduction of a novel translation initiation site, resulting in a shift of reading frame and effectively leading to haploinsufficiency. (**C**) Rare variants in the intronic sequence may lead—by different mechanisms—lead erroneous splicing, creating an out of frame transcript and resulting in haploinsufficiency. (**D**) Within the 3′UTR, multiple miRNA binding sites are present to regulate transcription. Rare variants in these sites may prevent miRNA binding and, thus, correct regulation of transcription. (**E**) Transcriptional regulation by enhancers and silencers is organized within topological-associated domains (TADs). Deletion of such a boundary may place genes under the control of ectopic enhancers and/or silencers of neighboring TADs, leading to dysregulation of gene expression.

**Figure 8 brainsci-13-00231-f008:**
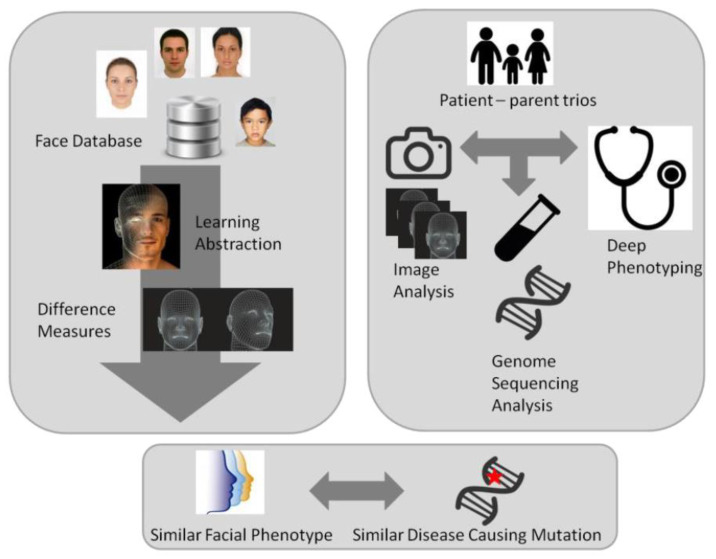
Novel strategy for characterizing novel gene mutations by combining the analysis of facial imaging algorithms with genomic data to establish a link between patients’ rare variants and potential facial gestalt.

## Data Availability

Not applicable.

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
