# Peer review of "The Genetics of Intellectual Disability"

_brainsci, 2023, doi:10.3390/brainsci13020231_

Round 1

Reviewer 1 Report

Jansen et al. summarize the current state of the genetics of intellectual disability in a well written and graphically well illustrated article. The review gives a good top level introduction to the topic.

I have no fundamental issues with the manuscript and only a few suggestions:

1. Page 3, first paragraph: Another benefit of a correct diagnosis is the possibility for patients and families to connect with other families, self-help groups or advocacy groups. Families might also gain access to special, disease-specific support programs. This could also be mentioned.

2. Page 5, subsection 3, last paragraph: This sentence reads awkwardly, I suggest rephrasing this like "While chromosome microarrays enabled genome-wide unbiased detection of structural rearrangements, similar methods for sequence variants at base-pair resolution were still lacking."

3. Page 5, subsection 4, last sentence: Also suggest rephrasing this, the meaning was not completely clear to me.

4. Page 6, first paragraph: It should be noted that long read NGS, not to mention long read WGS, is only now slowly starting to be used in routine diagnostics - I would say it is still the absolute exception.

5. Page 7, Figure 5: I think showing the yield of trio WES here for comparison (as the top tier test available in routine practice today) makes sense.

6. Page 9, subsection 6: With regards to variants unlikely to cause the phenotype, I would use something like "filtered out" rather than "declined".

7. Page 5, first paragraph, line 6: "the" is duplicated.

As a suggestion, the authors could also briefly mention the difficulty in diagnosing patients with more than one Mendelian disorder. This "double trouble" or "multiple molecular diagnoses" might affect as many as 5% of patients with genetic disorders. A phenotype-first approach in these patients is often hopeless.

Author Response

Thanks for the kind words regarding this review which is indeed intended to give a good level introduction to the topic.

Response to the Reviewer’s comments

  1. Page 3, first paragraph: Another benefit of a correct diagnosis is the possibility for patients and families to connect with other families, self-help groups or advocacy groups. Families might also gain access to special, disease-specific support programs. This could also be mentioned.

We have added the text suggestions to page 3 1st paragraph

  1. Page 5, subsection 3, last paragraph: This sentence reads awkwardly, I suggest rephrasing this like "While chromosome microarrays enabled genome-wide unbiased detection of structural rearrangements, similar methods for sequence variants at base-pair resolution were still lacking."

We removed this sentence and started in the next paragraph with: ‘While chromosome microarrays enabled genome-wide unbiased detection of structural rearrangements, similar methods for sequence variants at base-pair resolution were still lacking.’

  1. Page 5, subsection 4, last sentence: Also suggest rephrasing this, the meaning was not completely clear to me.

We have added  ‘… within one experiment’ to this sentence for clarification.

  1. Page 6, first paragraph: It should be noted that long read NGS, not to mention long read WGS, is only now slowly starting to be used in routine diagnostics - I would say it is still the absolute exception.

We have added the sentence: ‘Currently, WGS is only slowly starting to be used in routine diagnostics and therefore still an exception at the moment.’

  1. Page 7, Figure 5: I think showing the yield of trio WES here for comparison (as the top tier test available in routine practice today) makes sense.

Figure 5 has been changed into figure 6. In the text we have clarified figure 6 on page 6.

Note: figure 6 has been renumbered into figure 5.

  1. Page 9, subsection 6: With regards to variants unlikely to cause the phenotype, I would use something like "filtered out" rather than "declined".

‘declined’ has been replaced by ‘filtered out’ two times in the text on page 7 as suggested by the reviewer

  1. Page 5, first paragraph, line 6: "the" is duplicated.

one ‘the’ has been removed.

 As a suggestion, the authors could also briefly mention the difficulty in diagnosing patients with more than one Mendelian disorder. This "double trouble" or "multiple molecular diagnoses" might affect as many as 5% of patients with genetic disorders. A phenotype-first approach in these patients is often hopeless.

On suggestion of the reviewer, we have added the sentence: ‘Moreover, for diagnosing patients with more than one Mendelian disorder the phenotype-first approach would not work’ to page 4.

Reviewer 2 Report

This is a easy-to-read somehow superficial review of the recent history of how technical advancements have permitted to associate different forms of Intellectual Disability to specific genes or genomic regions.

I find that this review could be much more interesting if more specific examples are described and discussed. 

For example the difference in the type of results obtained by WES and WGS is described very superficially. Technically it incorrect to say that WES is done "to screen for mutations in known genes", a gene includes also the non-coding regulatory regions, maybe it might be better to say "to screen for coding variants of known genes". I think it might be very important to explain how WGS can identify regulatory variants associated to "quantitative" traits that can be part of the ID spectrum as well.

Figure 6 is not sufficiently described in the text, this figure is also an example for what is told later about facial analysis algoritms (Figure 7), but all is superficial and not well described. I find indeed interesting the facial analysis, it must be described better, possibly with some historical references and anecdotes it could be more fun for the reader.

Globally I think more examples and discussion should be included

Author Response

This is a easy-to-read somehow superficial review of the recent history of how technical advancements have permitted to associate different forms of Intellectual Disability to specific genes or genomic regions.

We underscore the remark by reviewer 1 who noted that ‘the review gives a good top level introduction to the topic.’

We intended to give just an introduction for a broad audience but added several parts to give it some more depth (see below and in the manuscript).

I find that this review could be much more interesting if more specific examples are described and discussed. 

For example the difference in the type of results obtained by WES and WGS is described very superficially. Technically it incorrect to say that WES is done "to screen for mutations in known genes", a gene includes also the non-coding regulatory regions, maybe it might be better to say "to screen for coding variants of known genes".

Thanks for this suggestion: we have changed this accordingly.

I think it might be very important to explain how WGS can identify regulatory variants associated to "quantitative" traits that can be part of the ID spectrum as well.

This is a good suggestion.

We have rewritten the part under ‘Challenges of NGS and beyond’ (starting on page 7) and changed the heading into ‘From exomes to genomes - from coding to non-coding variants’

We have also added a novel paragraph starting on page 8 ‘Genomes: from short reads to long reads and the incorporation of other -omics’

Figure 6 is not sufficiently described in the text, this figure is also an example for what is told later about facial analysis algoritms (Figure 7), but all is superficial and not well described.

We agree with the reviewer and added the following text to page 7 (Note that fig 6 has been renumbered into fig 5):

For most of these novel syndromes, the process to their first description is identical (Figure 5): after the identification of a first patient with NDD, a search for a larger series starts. This can be either through (inter)national collaborations and matchmaker exchange programs, or alternatively, by active (re)sequencing of patients with a similar NDD phenotype by targeted enrichment. For the latter, a popular approach has been the use of molecular inversion probes (MIPs)21,84-87, which is a high throughput and affordable technology to effectively sequence a large cohort of individuals on pathogenic variants in 5-50 genes simultaneously within a single experiment. Once multiple patients have been identified, more detailed further clinical characterization is performed using deep-phenotyping to established the phenotypic spectrum. Subsequently, depending on the function of the protein functional testing can be developed to prove a deleterious effect of the various gene variants. Also various disease models, either animal or cellular models, can contribute to the further molecular characterization of the gene/protein and its pathophysiological mechanism.  

I find indeed interesting the facial analysis, it must be described better, possibly with some historical references and anecdotes it could be more fun for the reader.

On the suggestion of the reviewer we added a novel paragraph under the heading ‘Quantïtative phenotyping’ have added the following to page 10:

The current state-of-the-art model is DeepGestalt 98, 99, demonstrating 60% accuracy in 216 different genetic syndromes. DeepGestalt however, like all other published studies since Ferry et al. in 2014, depends on proprietary, closed-source algorithms. With open science gaining traction in academia, sharing of code, models and data (when possible) is desirable, to ensure academic progress is not hindered, and ensure anyone can use these models, verify them and further build on and improve them. A novel AI-based phenomics approach has recently been presented by Dingemans et al who developed a PhenoScore based on the combination of facial recognition technology with Human Phenotype Ontology (HPO) data analysis.100 This novel open science approach, leading to this PhenoScore, allows not only to quantify rare disorders but genetic variation as well. 100  

Globally I think more examples and discussion should be included

Thanks for this suggestion. We have added several parts to enhance the depth and the impact of review without changing the overall concept.

Moreover have added figure 7 (and renumbered fig 7 into 8) to give an additional explanation for the techniques used and their outcome.

 We have also included several more references.